# Multiple Roles for Cytokines in Atopic Dermatitis: From Pathogenic Mediators to Endotype-Specific Biomarkers to Therapeutic Targets

**DOI:** 10.3390/ijms23052684

**Published:** 2022-02-28

**Authors:** Luca Fania, Gaia Moretta, Flaminia Antonelli, Enrico Scala, Damiano Abeni, Cristina Albanesi, Stefania Madonna

**Affiliations:** 1Integrated Center for Research in Atopic Dermatitis (CRI-DA), IDI-IRCCS, Via Monti di Creta, 104, 00167 Rome, Italy; l.fania@idi.it (L.F.); g.moretta@idi.it (G.M.); flamiantonelli@gmail.com (F.A.); e.scala@idi.it (E.S.); 2Clinical Epidemiology Unit, IDI-IRCCS, 00167 Rome, Italy; d.abeni@idi.it; 3Laboratory of Experimental Immunology, IDI-IRCCS, Via Monti di Creta, 104, 00167 Rome, Italy; s.madonna@idi.it

**Keywords:** atopic dermatitis, cytokines, intracellular pathways, endotypes, itch, biologics, small-molecule inhibitors

## Abstract

Atopic dermatitis (AD) is one of the most common chronic inflammatory skin diseases, which generally presents with intense itching and recurrent eczematous lesions. AD affects up to 20% of children and 10% of adults in high-income countries. The prevalence and incidence of AD have increased in recent years. The onset of AD mostly occurs in childhood, although in some cases AD may persist in adult life or even manifest in middle age (adult-onset AD). AD pathophysiology is made of a complex net, in which genetic background, skin barrier dysfunction, innate and adaptive immune responses, as well as itch contribute to disease development, progression, and chronicization. One of the most important features of AD is skin dehydration, which is mainly caused by filaggrin mutations that determine trans-epidermal water loss, pH alterations, and antigen penetration. In accordance with the “outside-inside” theory of AD pathogenesis, in a context of an altered epidermal barrier, antigens encounter epidermal antigen presentation cells (APCs), such as epidermal Langerhans cells and inflammatory epidermal dendritic cells, leading to their maturation and Th-2 cell-mediated inflammation. APCs also bear trimeric high-affinity receptors for immunoglobulin E (IgE), which induce IgE-mediated sensitizations as part of pathogenic mechanisms leading to AD. In this review, we discuss the role of cytokines in the pathogenesis of AD, considering patients with various clinical AD phenotypes. Moreover, we describe the cytokine patterns in patients with AD at different phases of the disease evolution, as well as in relation to different phenotypes/endotypes, including age, race, and intrinsic/extrinsic subtypes. We also discuss the outcomes of current biologics for AD, which corroborate the presence of multiple cytokine axes involved in the background of AD. A deep insight into the correlation between cytokine patterns and the related clinical forms of AD is a crucial step towards increasingly personalized, and therefore more efficient therapy.

## 1. Introduction

Atopic dermatitis (AD) is one of the most common chronic inflammatory skin diseases, characterized by intense itching and recurrent eczematous lesions [1]. The prevalence and incidence of AD have increased in recent years [1]. According to the Global Burden of Disease Study, the prevalence in high-income countries is between 15% and 20% among children, and up to 10% among adults [2,3]. No gender-related differences are reported, while for ethnicity a higher prevalence was observed in Afro-American U.S. children (19.3%) compared to Caucasians (16.8%) [1]. The onset of AD mostly occurs in childhood, usually followed by remission before adulthood, although some cases may reveal protracted disease activity (persistent AD) or even a primary outbreak in middle age (adult-onset AD) [4]. Several patients may also have an onset in childhood followed by a latency period with re-exacerbation in adulthood (chronic-relapsing form) or even, in a few cases, the first occurrence in adolescence (adolescent-onset AD).

AD is generally associated with other atopic diseases (atopic march), such as asthma, rhinitis, conjunctivitis, and food allergy, which further worsen quality of life [5,6]. Evidence of this association is provided by a study on Italian atopic schoolchildren [7], which reported a prevalence of asthma ranging from 46% in children with AD to 10% in unaffected ones, while the prevalence of allergic rhino-conjunctivitis in these population groups was 35.6% and 15.1%, respectively. Besides, atopic patients have a major risk of developing respiratory, contact, or food allergies.

AD clinically manifests as the evolution from an initial acute phase, characterized by pruritus and erythemato-vesicles/papules, to a chronic phase, during which skin appears more lichenified as a consequence of tissue remodeling and dermal fibrosis due to inflammation and scratching of the skin [1]. Acute and chronic lesions are often found in the same individual, often overlap, and clinically are sometimes difficult to distinguish [8]. Rarely, a simultaneous occurrence of psoriasis and AD may be observed in the same patient, making the differential diagnosis even more complex [9].

AD pathophysiology is made of a complex net, in which genetic background, skin barrier dysfunction, innate and adaptive immune responses, and itch contribute to development, progression, and chronicitation of disease [3]. One of the most important markers of AD is skin dehydration, which is mainly caused by *filaggrin* (*FLG*) gene mutations that determine trans-epidermal water loss, pH alterations, and antigen penetration [10]. In accordance with the “outside-inside” theory of AD pathogenesis, in a context of an altered epidermal barrier, antigens encounter epidermal antigen presentation cells (APCs), such as epidermal Langerhans cells and inflammatory epidermal dendritic cells (IEDC), leading to their maturation and Th2-mediated inflammation. APCs also bear trimeric high-affinity receptors for immunoglobulin E (IgE), and therefore IgE-mediated sensitizations play an important role as part of pathogenic mechanisms leading to AD.

In this review, we discuss the role of cytokines in the pathogenesis of AD considering patients with various clinical AD phenotypes. Moreover, we describe the cytokine patterns in patients with AD at different phases of the disease evolution, as well as in relation to the different phenotypes/endotypes, including age, race, and intrinsic/extrinsic subtypes. We also discuss the outcomes of current biologics for AD, which corroborate the concept that multiple cytokine axes act in the background of AD. A deep insight into the correlation between cytokine patterns and the related clinical forms of AD is the crucial step towards increasingly personalized, and therefore more efficient therapy.

## 2. Active Participation of Inflammatory Cytokines during AD Evolution

### 2.1. Role of Cytokines in Acute AD

AD disease is characterized by a biphasic inflammation, evolving from an initial, acute, Th2- and Th22-dominated phase to a chronic phase characterized by the concomitant presence of T helper (Th)1, Th2 cells, and Th17 cells [3]. Th2-derived cytokines, together with inflammatory mediators released by innate immune cells, such as mast cells, pathogenically contribute to the initiation and amplification of skin inflammation in AD lesions.

The role of the innate immune system in the early phase of AD has been demonstrated in experimental animal models [11,12] and is likely of clinical relevance in infancy [13].

A key player of innate immunity is the epidermal barrier and the loss-of-function *FLG* gene variants (R510X and 2282del4) that constitute a major predisposing factor for AD [14]. However, whether immune dysregulation results from skin barrier abnormalities, such as FLG lack, or it can be considered as the initial trigger leading to barrier deficiencies by downregulation of, for example, *FLG* gene expression, is still debated. A recent study indicates that *FLG*-deficiency renders skin equivalents more sensitive to the detrimental effects of IL-4 and IL-13 compared to skin equivalents with normal *FLG* expression, and therefore, defects in the epidermal barrier, skin permeability, and cutaneous innate immune response are not primarily linked to *FLG* gene deficiency but are rather secondarily induced by Th2 inflammation [15].

#### 2.1.1. IL-1 Cytokine Family

Regarding the innate immune responses, dysregulation of the IL-1 axis may account for the initiation of inflammatory responses in AD [16]. Indeed, an up-regulated expression of the IL-1-related cytokines IL-1β and IL-18 was observed in AD patients with *FLG* mutation. These cytokines promote lead to cutaneous inflammation through the induction of secondary cytokines, such as IL-8, and upregulation of endothelial adhesion molecules [16]. IL-1α is a pro-inflammatory cytokine released by keratinocytes after injury and by skin dysbiosis [17]. As one of the first and most important mediators in antigen presentation and induction of the inflammatory cascade, IL-1α has been considered as a therapeutic target in AD.

Similarly, IL-33, a cytokine structurally related to IL-1β and IL-18, is abundant in the epidermis of AD lesions [18,19]. However, it is unclear whether IL-33 is the cause or the result of AD. Of note, when up-regulated in keratinocytes of a transgenic mouse model, IL-33 induces severe eczema [20]. IL-33 is produced by endothelial cells and various epithelial cells, including keratinocytes, which constitutively express IL-33 as an inactive precursor [19]. In response to infection or tissue injury, IL-33 precursor is cleaved by caspase-1 to form an active secreted IL-33, which in turn activates mast cells, basophils, and group 2 innate lymphoid cells (ILC2) to secrete IL-4, IL-5, and IL-13 via the receptor suppression of tumorigenicity 2 (ST2) [21]. Other than triggering Th2 polarization, IL-33 promotes the secretion of pruritic cytokines, including TSLP and IL-31, from keratinocytes and Th2 cells, respectively, which amplifies Th2 responses [21,22]. IL-33 also mediates the itch response by activating itch-sensing sensory neurons [19] and contributes to the disruption of the epidermal barrier function via the down-regulation of FLG and claudin-1 levels [23] (Figure 1).

Danger signals from barrier disruption and microbial invasion trigger the production of additional keratinocyte-derived cytokines, such as IL-6, IL-23, and tumor necrosis factor (TNF)-α, which exhibit pro-inflammatory activities and promote the activation, differentiation, and recruitment of inflammatory cells to skin lesions [24,25,26,27]. The combination of IL-1β and IL-6, together with transforming growth factor-β (TGF-β), promotes Th17 cell activation, which plays an important part in early stages of AD [28].

IL-36 cytokines are other innate immunity players belonging to the IL-1 family that are upregulated in the skin of acute and chronic AD. Interestingly, colonization with *S. aureus* in a murine AD model induces inflammation through IL-36R- and IL-1R-dependent signaling [29].

In acute phases of AD, keratinocytes in barrier-disrupted epidermis also produce large amounts of thymic stromal lymphopoietin (TSLP) and interleukin (IL)-25, which promote Th2 immune deviation via OXO40L/OX40 signaling [30].

#### 2.1.2. TSLP

TSLP is an epithelial cell-derived IL-7-like cytokine that is released in response to mechanical injury, microbial infection, and allergen exposure [21]. Increased expression of TSLP was observed in *FLG*-depleted keratinocytes after TLR-3, TLR-5, and TLR-2/-6 ligand stimulation [30]. The expression of TSLP in AD skin correlates with the severity of the disease and the degree of epidermal barrier disruption [31]. TSLP directly activates dendritic cells to polarize naive T cells towards Th2 cells that secrete IL-4, IL-5, and IL-13, which further induce TSLP release by keratinocytes themselves [32,33].

TSLP also inhibits the production of antimicrobial peptides by keratinocytes, such as hBD-2, which correlates with the susceptibility of the skin to infections [34]. Furthermore, keratinocyte-derived TSLP contributes to pruritus induction by binding TSLP receptors located on cutaneous sensory neurons [35] (Figure 1).

#### 2.1.3. IL-17 Cytokine Family

IL-25 is another key mediator of the development of Th2 response [36]. IL-25, also known as IL-17E, is a member of the IL-17 family, sharing structural similarity with other IL-17 members [36,37]. Although produced by several types of immune cells, such as allergen-activated mast cells, eosinophils, basophils, dermal dendritic cells, T cells, or ILC2, a vast amount of IL-25 derives from epithelial cells and in particular, by keratinocytes [37,38]. An increased number of IL-25-expressing epidermal keratinocytes was detected in lesional and non-lesional AD skin compared to the skin from healthy individuals [38]. IL-25 is known to induce Th2 cells via direct activation of naive CD4^+^ T cells or ILC2 activation to promote innate type-2 immune responses [37] (Figure 1).

#### 2.1.4. Th2-Derived Cytokines

AD is generally considered a Th2-mediated disease. In the acute phase, Th2 lymphocytes mount strong inflammatory responses with the secretion of type-2 cytokines, such as IL-4 and IL-13, which downregulate the levels of FLG, loricrin (LOR), and involucrin (INV) in keratinocytes, and exacerbates epidermal barrier dysfunction [10]. Keratinocytes constitutively express functional IL-4 and IL-13 receptors [39] and produce the eosinophil chemokine CCL26 in response to IL-4 and IL-13. Other than regulating IgE antibody production in B cells, IL-4 and IL-13 have been reported to act directly on itch-sensory neurons to promote itch [40,41]. Th2 cells also secrete IL-31, a pruritogenic cytokine described below (see Section 2.2).

#### 2.1.5. IL-22

In addition to the strong Th2 activation, the acute AD lesions in adults are also characterized by a Th22 response, with release of IL-22 and S100A proteins [42,43]. Following scratching, the endogenous TLR-4 ligands stimulate the production of IL-23 from keratinocytes [44]. IL-23 in turn activates IL-23R-expressing DCs, which trigger Th22 immune response mediated by the aryl-hydrocarbon receptor (AHR) [45]. The robust IL-22 expression results in epidermal hyperplasia and barrier defects of affected skin [46] (Figure 1).

### 2.2. Role of Cytokines in Chronic AD

Although a Th2 signature predominates in the acute phase, a Th2 towards Th1 switch has been long deemed to promote disease chronicity [46,47].

More recent findings demonstrated that the progression of acute-to-chronic AD is associated with quantitative rather than qualitative changes in cytokine responses, with the intensification of Th2, Th1, and Th17 responses in chronic inflammation of AD [47]. Th17-related responses lead to the accumulation of IL-17A and IL-17F cytokines in chronic AD lesions. It is yet to be elucidated whether IL-17 plays a critical role in AD as it does in psoriasis.

In chronic AD, lymphocyte-released cytokines synergize, thus amplifying the inflammatory responses. For example, in keratinocytes, IL-4 potentiates the action of IFN-γ and TNF-α in inducing CXCR3 agonistic chemokines, such as CXCL9, CXCL10, and CXCL11, which recruit more T cells into inflamed skin [48].

The shift to chronic AD is also accompanied by increased activity of IL-36 cytokines [47]. IL-36 cytokines are expressed predominantly by epidermal keratinocytes and act on many cells including endothelial and immune cells [49,50,51]. IL-36 cytokines do not act directly on T cells but instead can stimulate maturation and function of DCs and through them drive T cell proliferation, thereby propagating and amplifying immune responses in the skin [52]. Recently, Shao et al. demonstrated that the serine-threonine kinase IRAK2 is the main intracellular effector of IL-36 and IL-1 cytokines in human keratinocytes, and its levels correlate with disease severity in AD and psoriasis [53] (Figure 1).

Chronic itch is the major symptom in AD patients [54,55]. Scratching exacerbates predisposing dermatitis, which may further enhance pruritus and result in an itch/scratch vicious cycle typically occurring in chronic AD [56].

Other than IL-33 and TLSP, IL-31 has also a pruritogenic activity through activation of the heterodimeric receptor IL-31 receptor A (IL31RA)/Oncostatin M receptor (OSMRβ), expressed by dorsal root ganglia neurons, keratinocytes, and various innate immune cells [56]. IL-31 is a cytokine produced by various cells including Th2 cells, macrophages, dendritic cells, and eosinophils, and its levels were increased in lesional and non-lesional skin of AD patients [57,58,59]. In parallel, serum levels of IL-31 correlated with disease severity in AD patients [60,61]. Finally, in human AD skin, keratinocytes show elevated levels of IL31RA/OSMRβ expression resulting in stronger receptiveness to IL-31 [62]. In addition to its pruritogenic function, IL-31 is known to directly inhibit the differentiation of keratinocytes by downregulating the expression of barrier/differentiation-related proteins, which results in the disruption of epidermal barrier function [63,64,65] (Figure 1).

A summary of the main cytokines involved in AD pathogenesis is reported in Table 1.

## 3. Cytokines as Endotype-Specific Biomarkers

AD is a heterogeneous disease with various clinical manifestations (phenotypes) sustained by specific molecular mechanisms (endotypes). Several endotypes based on the age of onset [66,67], ethnic origin [68], and clinical features can be distinguished [69,70].

### 3.1. Age of AD Onset

Regarding the age of onset, early-onset (infantile < 2 years, childhood 2–12 years, adolescent 12–18 years) and adult onset (in patients aged > 18) can be distinguished. Recently, a specific AD subtype has been identified to include elderly onset AD in patients aged > 60 years.

Together with Th2 (IL-13, IL-31 and CCL17) activation, enhanced Th22 (IL-22 and S100As), Th17 (IL-17A, IL-19, CCL20, LL37 and peptidase inhibitor 3/elafin), and Th1 (IFN-γ and CXCL9/CXCL10/CXCL11) pathways characterize adult AD, whereas pediatric patients exhibit lower Th1 activation but a higher expression of Th9 (IL-9), and innate markers (IL-1β, IL-8 and IFN-α1), as well as dysfunctions in epidermal lipid metabolism responsible for the barrier alterations [71,72]. A recent study on tape strips from early-onset pediatric AD highlighted abnormalities in Th2-, Th22-, and Th17-related pathways also in non-lesional skin [73]. Non-lesional skin in pediatric patients with AD showed higher levels also of IL-19 and LL37, as well as of epidermal proliferation (Keratin 16 and S100As) markers [71] (Figure 2).

An attempt to stratify adult patients with AD by the serum biological markers has succeeded in identifying four clusters [74]. In this study, sera from 193 adult patients with moderate AD, severe AD, and healthy control subjects without AD were analyzed for serum mediators, total IgE levels, and allergen-specific IgE levels. A principal component analysis yielded four distinct clusters of patients with AD. Cluster 1 showed more severe clinical scores and more affected areas enriched with Th2 cytokines (IL-13, IL-5), as well as IL-22 and IL-33, with the highest levels of TARC, pulmonary and activation-regulated chemokines (PARC), tissue inhibitor of metalloproteinases 1 (TIMP1), and soluble CD14. Cluster 2 was characterized by a relatively low inflammatory state particularly distinctive from that of the other clusters by virtue of having low serum levels of Th2/severity-related (MDC, PARC, and TARC) and eosinophil-related (RANTES, eotaxin, and eotaxin-3) markers. Cluster 3 had more severe clinical scores with the lowest levels of IFN-β, IL-1, and epithelial cytokines, such as TSLP. Cluster 4 had milder clinical scores but the highest levels of the inflammatory Th2-related (IL-4, IL-5, and IL-13), Th1-related (IFN-γ, TNF-α, and TNF-ß), Th17-related (IL-17 and IL-21), and epithelial-related (IL-25, IL-33, and TSLP) cytokines (Figure 2). This study has been recently confirmed by the same authors in a different cohort of patients with severe AD [75].

Of note, Lauffer et al. recently identified three distinct endotypes based on serum cytokines and clinical features in children with AD that discriminate a persistent course from remission [76].

### 3.2. Ethnicity and AD

In relation to race, although Th2 markers (IL-13, CCL17, CCL18, and CCL22) were similar between Asian and European American patients with AD, Th1 markers were significantly lower in lesional and non-lesional tissues of Asian patients with AD [66]. Asian patients have accentuated polarity of the Th22 (IL-22 and S100A12)/Th17 (IL-17A and the related CCL20 marker) pathways, and also exhibit epidermal barrier defects despite relative maintenance of *FLG* and *loricrin* gene expression [77]. In contrast, African American patients do not exhibit *FLG* mutations and have distinct attenuation of Th17/Th1 axes [78,79]. Levels of IL-19, which are induced by IL-4, IL-13, and IL-17, and augment IL-17-dependent effects on keratinocytes [80,81] were significantly greater in AD lesions of Asian versus EA patients (Figure 2).

### 3.3. Intrinsic and Extrinsic AD

The endotype pattern of AD also includes extrinsic and intrinsic AD [79,82]. Extrinsic (allergic) AD represents about 80% of adult atopic patients and is associated with a high level of serum IgE. Elderly patients with this AD endotype show frequent allergic sensitization to airborne allergens and to food allergens [83].

The intrinsic (non-allergic) AD is a less common subtype (≈20%); however, it affects the elderly in an increased proportion [84]. Intrinsic AD is characterized by normal or low serum IgE levels, the absence of atopic background, and a lack of sensitization to environmental allergens. However, specific IgE against enterotoxins of *S. aureus* and other microbial antigens have been identified [84,85]. Intrinsic AD shows similar Th2 but higher Th17, Th1, and Th22 immune activation compared with extrinsic AD, with higher amounts of the inflammatory IL-17, CXCL8, IFN-γ, and IL-22 cytokines and related chemokines in intrinsic AD lesional skin (79) (Figure 2).

Three main phenotypes of AD have been proposed in a recent PRACTALL document: non-lesional skin, acute disease flares, and chronic remitting relapsing AD. A type 2 immune response is present in all three phenotypes, with a peak in acute disease flares. Here, the type-2 immune response is associated to a Th22- and Th17-driven inflammation present in non-lesional skin, whereas Th22- and Th1-driven responses are abundant in patients with the chronic form of AD. Epithelial dysfunction is present in non-lesional skin and in patients with chronic AD [86].

In conclusion, stratification of AD patients into distinct cytokine-based endotypes might contribute to more personalized medicine and may be important to better inform which patients are most likely to benefit from specific targeted therapies and to design disease-modifying strategies. Some biomarkers, such as CCL17 chemokine, are a consistent measurement of AD severity in multiple clinical trials. However, multiple biomarkers will probably be needed as a signature profile in AD to predict the severity, comorbidities, and treatment response.

## 4. Involvement of Th1/Th2-Derived Cytokines in AD Comorbidities

Several studies have considered AD as a systemic disorder, due to its association to a variety of conditions, known as atopic comorbidities. They include infectious, atopic, autoimmune, cardiovascular, and psychiatric disorders, which share similar immune pathogenic mechanisms with AD [87]. Frequent and severe bacterial and viral cutaneous infections are typical comorbidities of AD. *S. aureus* is found in up to 90% of AD lesions, while in healthy individuals, a 5–30% prevalence of colonization is reported [88]. This agent seems to be involved in the pathogenesis of AD through the induction of IL-4, IL-13, and IL-22 [87]. The most common viral complication in AD patients is eczema herpeticum (EH), which tends to be more frequent in individuals with evident Th2 polarization, and consistent allergen sensitization. Th2/Th1 impaired balance leads to lower levels of antimicrobial peptides and barrier proteins. AD patients affected by EH also display low IFN-γ levels and a down-regulation of IFN-γ receptors, resulting in defective responses to viral multiplication [88].

The skin of individuals with AD clearly shows greater vulnerability to bacterial or fungal colonization. The predominant skin infection in AD is caused by *S. aureus* and the presence of specific IgE antibodies to *Staphylococcus* exotoxins (SE) has been demonstrated in patients with AD [89]. Moreover, patients with moderate to severe AD appear to have IgE to bacterial antigens more frequently than patients with mild symptoms. Furthermore, these exotoxins may play a central role not only as “allergens”, but also as “superantigens”, through the restricted non-MHC activation of T cells bearing the reactive TCRVβ family. *S. aureus* is in fact able to generate a series of exotoxins with super-antigenic capacity that can stimulate specific subsets of T lymphocytes. As a consequence, treatment of *S. aureus* skin infections with specific antibiotics significantly reduces the clinical severity of the disease [90].

Fungi are common and important allergens in the environment, and fungal sensitization to *C. albicans*, *A. alternata*, T., *P. chrysogenum*, *A. fumigatus*, and *M. furfur* is often observed in AD. IgE sensitization to *Malassezia* is observed in AD and not in patients with allergic rhinitis or asthma without AD. The presence in AD of specific IgE recognition of the manganese superoxide dismutase (MnSOD), a protein probably involved in IgE-mediated self-reactivity, has been demonstrated [91,92]. The hypothesis of a molecular mimicry followed by cross reactivity was raised by Schmid-Grendelmeier et al., as a result of the primary sensitization to MnSOD belonging to the *Malassezia sympodialis*. IgE-mediated reactivity against self-proteins with structural similarity to exogenous allergens has been hypothesized as a further potential mechanism involved in AD pathogenesis [93]. In addition to exogenous allergens, a specific IgE response can be directed against a range of human proteins located in a variety of cell and tissue types [94]. In the last two decades, five IgE-reactive auto allergens were identified using sera from severe AD patients on a human cDNA library [95,96,97,98]. A significant correlation was observed between IgE self-reactivity and AD severity. A reduction in the IgE response was observed after treatment with cyclosporine, suggesting a role for self-reactive IgE as a marker of tissue damage.

The susceptibility to skin infections is caused, at least in part, by the reduced expression of the host defense peptides (HDPs) LL-37, hBD-2, and hBD-3 observed in AD patients [99,100]. These peptides exhibit anti-microbial activities against skin pathogens, pro- and anti-inflammatory properties, and immunomodulatory activities [101,102]. In particular, HDPs induce cytokine and chemokine production and promote cell proliferation and migration [101,102]. HDPs also mediate the maintenance of epithelial barrier function by regulating the trans-epidermal water loss and distribution of tight junction proteins [103].

The reduced levels of HDPs may be explained by the predominance of Th2-derived cytokines, which act as strong inhibitors of LL-37, hBD-2, and hBD-3 production [102,103]. The pruritic cytokines, including IL-31 and TSLP, also have inhibitory effects on the production of these anti-microbial peptides [104,105]. *FLG* gene mutation in AD skin promotes perturbation of the skin barrier, but the impact of FLG on hBD levels remains controversial [99].

Atopic comorbidities, such as allergic rhinitis, asthma, and food allergy, are considered as the major criteria for AD diagnosis. They share with AD the Th2 polarization of lymphocytes, regardless of the presence or absence of IgE. Even the eyes can be frequently affected by the atopic condition, which may cause keratoconjunctivitis. Of note, the prevalence of hand eczema and allergic contact dermatitis (ACD) is increased in patients with AD [106], probably due to the action of the Th2-related (IL-4, IL-13, IL-25) and Th22-related (IL-22) cytokines, which reduce FLG levels in keratinocytes. The subsequent barrier dysfunction allows penetration of irritants and contact allergens. In addition, AD and ACD appear to share some immune pathways, including Th1, Th2, Th9, and Th17 [107,108].

Moreover, Koga C. et al. demonstrated a positive correlation of Th17 levels with AD clinical severity [109], whereas Nakae S. et al. reported that ACD reaction is weaker in the absence of IL-17 [110].

AD patients often use topical products, and these products may contain substances that lead to contact sensitization [111]. The last suggested mechanism to explain the association between AD and ACD is the skin bacterial colonization detected in most AD patients, which may create an inflammatory environment favoring contact sensitization [112]. The risk of irritative contact dermatitis is approximately three-fold in AD patients due to the impaired barrier function. The altered function of FLG is a common pathogenic mechanism in AD, and this can also be observed in ichthyosis vulgaris and keratosis pilaris. Indeed, an association of these diseases with AD has been reported [106].

Furthermore, the increased Th17 activity could, at least in part, explain the association between AD and some autoimmune diseases (i.e., inflammatory bowel diseases, rheumatoid arthritis, and lupus erythematosus) [113]. The correlation between AD and autoimmune diseases, including alopecia areata and vitiligo, could be due to the presence of some specific susceptibility genes shared among them.

In addition, AD patients more frequently suffer from depression and suicidality [114]. According to some studies, they also are at an increased risk of developing attention deficit hyperactivity disorder during childhood [115]. This could be explained by the relevant impact of AD on the patients’ quality of life [116], and more specifically by the prevalence and intensity of itching often associated with sleep disturbances, which may interfere with brain development in children [117], but also by early exposure of the central nervous system to inflammatory Th2 cytokines and systemic corticosteroids in pediatric age [118].

An increased risk of non-Hodgkin lymphomas has been observed in AD patients [119], possibly due to chronic inflammation and Th2 polarization able to reduce Th1 anti-neoplastic activity. An increased cardiovascular risk has even been found with a higher incidence of myocardial infarction and congestive heart disease; however, this seems to be attributable to poor health behaviors and more frequent cardiovascular risk factors rather than the atopic condition itself [120].

## 5. Cytokines and Their Intracellular Effectors as Therapeutic Targets in AD

Therapeutic strategies for AD vary according to several factors, including the severity of disease, extent and location of the affected body area, age, comorbidities, and quality of life of the patient. In recent years, the therapeutic choice for moderate to severe AD has been widened by the remarkable development of new target therapies, which not only allow more personalized treatment, but also bypass the possible toxicities of conventional therapies [121,122]. Some of these drugs are still in the trial phase, but others are already approved by the FDA and EMA (Table 2).

Dupilumab is a human monoclonal antibody directed against subunit α of the IL-4 receptor, which is also part of the IL-13 receptor. Through inhibition of the signaling of these two cytokines, dupilumab significantly improves Investigator’s Global Assessment (IGA), EASI (Eczema Area and Severity Index), and symptoms [123]. The most frequently reported adverse event in clinical trials in AD patients was conjunctivitis (8–22% of cases), but, in patients without AD and treated with dupilumab, no conjunctivitis occurred, making this a disease-specific adverse effect [124]. The safety and efficacy of dupilumab allowed its approval in 2017 in the U.S. and in Europe for moderate-severe AD in adults [124] and its recent approval even for adolescents from 12 years of age [125]. To date, it is not clear how long the therapy should be continued or whether or not it can be alternated with other therapeutic approaches.

The IL-13 signaling pathway is also blocked by other monoclonal antibodies currently being tested, in particular tralokinumab and lebrikizumab, which showed excellent results in clinical trials. Specifically, tralokinumab blocks the binding of IL-13 to both receptor chains, whereas lebrikizumab only blocks binding to the α1 chain, conversely allowing binding to the α2 chain, which plays a useful regulatory role [123] (Table 2).

Even the IL-31 pathway is currently under investigation. IL-31R seems to play a key role in the modulation of pruritus in AD patients. Indeed, it is expressed not only by immune cells, but also by keratinocytes and cutaneous peripheral sensory neurons. Nemolizumab is a humanized antibody targeting α subunit of IL-31R that has shown significant improvements in AD severity scores with few adverse effects (i.e., nasopharyngitis and upper respiratory infections) [123]. There is no approved indication for this drug yet [124]. Therapeutic strategies to inhibit IL-33 activity are under investigation as treatments for moderate to severe AD.

An anti-IL-33 monoclonal antibody, etokimab, was used in recent phase 2 clinical trials, showing an important reduction in skin inflammatory cascades [126].

Other signaling pathways under investigation are TSLP, OX40, and IL-22. TSLP activates dendritic cells, inducing Th2 polarization; it is the target of the human monoclonal antibody tezepelumab, which shows positive results in trials. OX40 is a costimulatory molecule of the TNF receptor, expressed mainly on T lymphocytes. Since the OX40 ligand is overexpressed on dendritic cells in AD patients, blocking this pathway is the new target of monoclonal antibodies GBR830 and KHK4083. IL-22 is a major contributor to cytokine-mediated dysfunction in keratinocytes and endothelial cells in AD. Fezakinumb, an IL-22 antagonist, has been developed, demonstrating good outcomes in phase 2 studies [123] (Table 2).

In addition to biological drugs, small-molecule inhibitors are a useful pharmacological class in development for the treatment of AD. Some of these drugs directly block JAK/STAT proteins, which are intracellularly activated by cytokine receptors and shared by groups of cytokine signals, in particular those related to IL-4, IL-5, IL-13, IL-31 and TSLP. For example, JAK1 is shared by signaling pathways induced by IL-4, TLSP, IL-13, IL-22 and IFN-γ, whereas TYK2 is common to IL-13 and IL-22 signaling. Based on these molecular features, most of JAK inhibitors synergistically act as inhibitors of both the inflammatory responses and pruritus [127].

Tofacitinib, a JAK1/JAK3 inhibitor, has been approved for RA and psoriatic arthritis, whereas baricitinib is a JAK1/JAK2 inhibitor approved for RA and the treatment of AD in Europe. Recently, baricitinib received emergency approval, in combination with the antiviral remdesivir, for the treatment of COVID-19 [128]. Peficitinib is a pan-JAK inhibitor approved in Japan for the treatment of RA, whereas delgocitinib (JTE-052) is a topical JAK inhibitor approved in Japan for AD [129]. Finally, upadacitinib has a degree of selectivity for JAK1 over JAK2 and is approved for the treatment of RA and AD in Europe [130].

The systemic JAK inhibitors (baricitinib, upadacitinib, and abrocitinib) show a rapid efficacy with rapid improvement of pruritus and a good tolerability in AD patients. However, topical application of JAK inhibitors is expected to play an important role in new therapeutic concepts for AD patient management, particularly for children (Table 2).

## 6. Conclusions

In summary, the complex clinical heterogeneity of AD needs newer and more effective treatments that are able to control the disease and improve the quality of life of patients. To date, biologics have showed long-term control of AD symptoms, whereas JAK inhibitors provide rapid relief in pruritus and inflammation. However, although well-tolerated, benefit–risk ratio of JAKs inhibitors remains a key issue for pharmacovigilance.

Some of these drugs even have the potential to modify the disease and could impact the atopic march and other comorbidities, if applied in the early phase of AD. Of note, specific cytokine profiles and barrier abnormalities characterize different AD groups and should be carefully considered when future therapeutic agents are being developed or tested for younger patients with AD. Therefore, accompanied by a deep characterization of different phenotype and endotype subsets, the application of precision medicine could provide new prospects for the optimal treatment of AD.

## Figures and Tables

**Figure 1 ijms-23-02684-f001:**
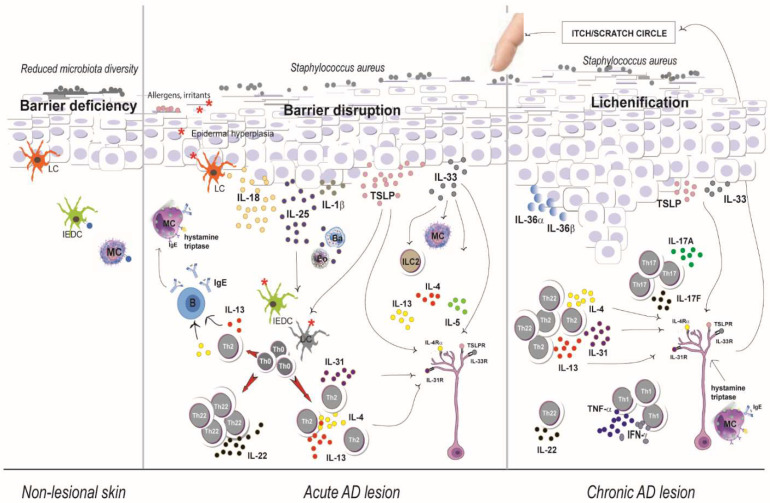
A simplistic overview of AD pathogenesis. Non-lesional skin has an epidermal barrier deficiency with a reduced diversity of the microbiome. In acute AD lesion, Langerhans cells, IEDC bearing specific IgE bound to the high affinity receptor for IgE, and dermal dendritic cells bind allergens and antigens. Keratinocyte-derived (IL-18, IL-1β, IL-33, TSLP and IL-25) and Th2-cell-derived cytokines IL-4, IL-13, and IL-31 directly activate sensory nerves, which promotes pruritus. During transition to chronicity, itch is amplified by various pruritogens (e.g., antigens and molecular mediators such as histamine and other substances). Scratching exacerbates dermatitis, which may further enhance pruritus and result in an itch/scratch vicious cycle. Chronic AD lesional skin is characterized by the intensification of Th2, Th1, and Th17 responses. DC = dendritic cell, MC = mast cell, IDEC = inflammatory dendritic epidermal cell, ILC = innate lymphoid cell. LC = lymphoid cell, B = B cell, Eo = eosinophile, Ba = basophile, IFN = interferon, IL = interleukin, Th = T-helper cell, Th0 = naive T cell, TNF = tumor necrosis factor, TSLP = thymic stromal lymphopoietin.

**Figure 2 ijms-23-02684-f002:**
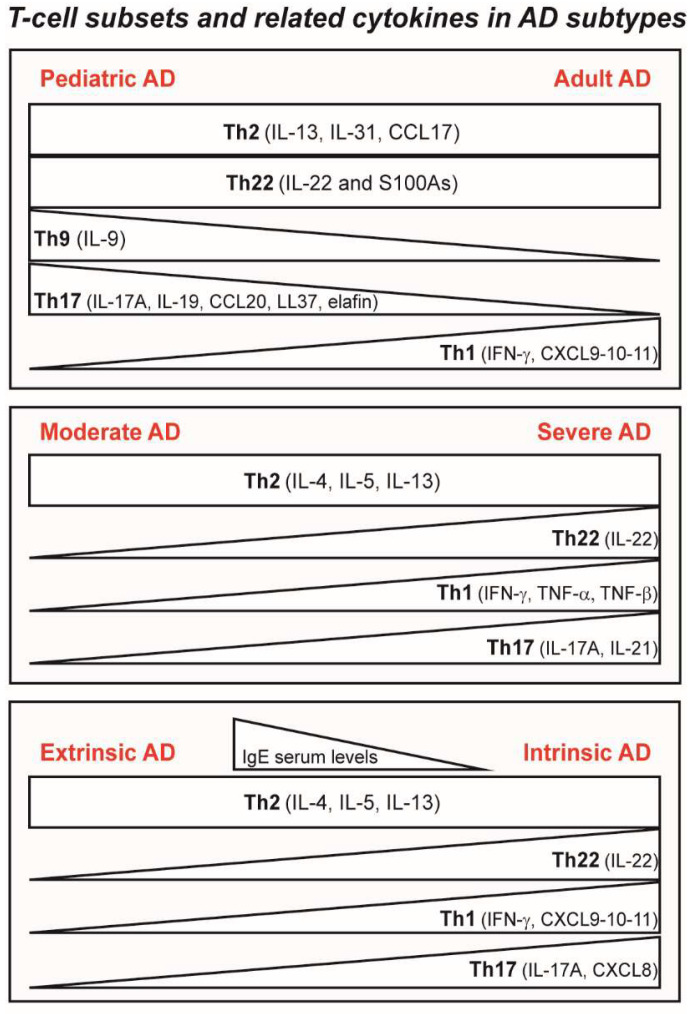
A schematic picture of the T cell subsets and cytokine profiles in the main AD endotypes. Adult AD is characterized by the intensification of Th1 signature, whereas pediatric AD shows the predominance of Th17(IL-17A, IL-19, CCL20, LL-37, elafin) and Th9 (IL-9)-related cytokines. Moderate and severe AD share Th2-dependent immune responses, with the identification of Th22 (IL-22), Th1 (IFN-γ, TNF-a, TNF-β), and Th17 (IL-17A and IL-21)-related cytokines in severe AD. Patients with extrinsic AD tend to exhibit barrier disruption, which causes repeated allergen exposure and B-cell activation resulting in hyper-IgE levels in serum. The cytokine spectrum of patients with intrinsic AD is further complicated with additional cytokine axes, including Th1 (IFN-γ, CXCL9, 10, 11) and Th17 (IL-17A, CXCL8).

**Table 1 ijms-23-02684-t001:** Key biological effects of the main cytokines in AD pathogenesis.

Cytokine	AD Phase	Function	Refs
IL-1α, IL-1β (IL-1 family)	Acute	Promote the recruitment of leukocytes and regulate synthesis of the extracellular lipid bilayers.	[16,17]
IL-33 (IL-1 family)	Acute	Activates mast cells, basophils, and ILC2; promotes the secretion of pruritic cytokines, i.e., TSLP and IL-31, from keratinocytes and Th2 cells, respectively.	[18,19,20,21,22,23]
TSLP	Acute/Chronic	Activates dendritic cells to polarize naive T cells towards Th2 cells; induces pruritus by binding TSLP receptors on cutaneous sensory neurons	[30,31,32,33,34,35]
IL-25 (IL-17 family)	Acute/Chronic	Induces innate and adaptive immune responses by activating ILC2 or polarizing naive T cells to Th2 cells.	[36,37]
IL-4/IL-13 (Th2 cytokines)	Acute/Chronic	Exacerbate epidermal barrier dysfunction; regulates IgE antibody production in B cells; promotes itch directly acting on sensory neurons.	[10,40,41]
IL-22	Acute	Induces epidermal hyperplasia and barrier defects of affected skin	[42,43,44,45,46]
IL-17A, IL-17F (IL-17 family)	Chronic	It is yet to be elucidated whether IL-17 plays a critical role in AD.	[46,47]
IFN-γ/TNF-α (Th1 cytokines)	Chronic	Induce CXCR3 agonistic chemokines, which recruit more T cells into inflamed skin.	[48]
IL-36s (IL-1 family)	Acute/Chronic	Levels of IL-36 cytokines correlate with disease severity in AD.	[52,53]
IL-33	Acute/Chronic	Has pruritogenic activity; inhibits keratinocyte differentiation by downregulating the expression of barrier/differentiation-related proteins.	[56,57,58,59,60,61,62,63,64,65]

**Table 2 ijms-23-02684-t002:** Therapeutic approaches in AD: molecular and cellular targets of biologics and small molecules.

**Biologics**	**Molecular Target**	**Intracellular Mediators**	**Cellular Target**
Dupilumab	IL-4/IL-13 receptor (α subunit)	JAK1/JAK3	Keratinocytes, B cells, cutaneous peripheral sensory neurons
Tralokinumab	IL-13 receptor (α1 and α2 chains)	JAK1/JAK2	Keratinocytes, B cells, cutaneous peripheral sensory neurons
Lebrikizumab	IL-13 receptor (α1 chain)	JAK1/JAK2	Keratinocytes, B cells
Nemolizumab	IL-31 receptor (α subunit)	JAK1/JAK2	Immune cells, keratinocytes and cutaneous peripheral sensory neurons
Etokimab	IL-33	Myd88/IRAK-1/IRAK-4	Keratinocytes, cutaneous peripheral sensory
GBR830 and KHK4083	OX40	TRAF-5, -6, -2	Th2 cells
Tezepelumab	TSLP	JAK1, JAK2	DC, ILC2, MC, sensory neurons, Ag-specific Th2 cells
Fezakinumb	IL-22	TYK2, JAK1	Keratinocytes, dermal endothelial cells
**Small Molecules**	**Intracellular Target**	**Cytokine Target**	**Clinical Trial Stage**
Tofacitinib	JAK1/JAK3	IL-4, partly IL-13, TSLP, IL-22, IL-31	Phase II completed (Topical) (NTC#02001181)
Baricitinib	JAK1/JAK2	IL-13, TSLP, partly IL-4, IL-31, IL-22	Phase III completed (NTC#03559270)
Peficitinib	Pan-JAK	IL-4, IL-13, IL-31, TSLP, IL-22	Phase II ongoing (NTC#04218877)
Delgocitinib	Pan-JAK	IL-4, IL-13, IL-31, TSLP, IL-22	Phase II ongoing (NTC#03725722)
Upadacitinib	JAK1	Partly IL-4, IL-13, TSLP	Phase III ongoing (NTC#03569293)
Abrocitinib	JAK1	Partly IL-4, IL-13, TSLP	Phase III completed (NTC#04564755)

## Data Availability

Not applicable.

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
