# Peer review of "Multiple Roles for Cytokines in Atopic Dermatitis: From Pathogenic Mediators to Endotype-Specific Biomarkers to Therapeutic Targets"

_ijms, 2022, doi:10.3390/ijms23052684_

Round 1

Reviewer 1 Report

In the paper, the authors discuss the role of cytokines in the pathogenesis of AD. Cytokine patterns in patients with AD at different phases of the disease, as well as in relation to different phenotypes, such as age, race, and subtypes are discussed. The current biologics used for AD and targeting cytokines are also covered.

Suggestions for improvement

-The authors are encouraged to divide paragraph 2 in subparagraphs dedicated to the different cytokine families.

-Tables summarizing the involvement of the different cytokine families in AD should also be provided

Author Response

As suggested by this Reviewer, we divided paragraph 2.1 into five subparagraphs (2.1.1-2.1.5) dedicated to specific cytokine families.

Consistently with the Reviewer's request, we have added a Tabble (Table 1) summarizing the main functions of the cytokine families in AD pathogenesis

All corrections or changes have been indicated in red.

Reviewer 2 Report

Albanesi et al. reviewed the pathophysiology of atopic dermatitis (AD), a common inflammatory skin disorder with emerging targeted treatment measures with remarkable efficacy. Understanding the complex pathophysiology of AD is crucial in uncovering skin biology and could be a key for successful treatment in the era of personalized medicine that we will experience near future.

From this standpoint, they adroitly summed up AD pathophysiology, particularly focusing on cytokine networks that presumably cross-talk each other. This manuscript is well-written, easy to read, and will add valuable information for novice and experienced clinical scientists interested in this trait. 

No major concerns are raised by this reviewer, remaining a few minor issues.

1. Line 375~ “Recently, IL-9 levels----absence of IL-17”.: This sentence seems repetitive since the authors stated the role of Th17 in the early phase clearly (L141) and might be revised or deleted for further clarification.

2. Table 1: Please add clinical trial numbers (NCT# from ClinicalTrials.gov).

Author Response

We corrected the sentence starting in line 375 which has erroneously repeated, as indicated by the Reviewer

As suggested by this Reviewer, the number of Clinical trial studies have added in Table 1 (now Table 2 for the insertion of the new Table 1)

All corrections or changes have been indicated in red

Round 2

Reviewer 1 Report

The authors have addressed the points previously risen